# Out-of-Domain Intent Detection Considering Multi-turn Dialogue Contexts

## Abstract

Out-of-Domain (OOD) intent detection is vital for practical dialogue systems, and it usually requires considering multi-turn dialogue contexts. However, most previous OOD intent detection approaches are limited to single dialogue turns. In this paper, we introduce a context-aware OOD intent detection (Caro) framework to model multi-turn contexts in OOD intent detection tasks. Specifically, we follow the information bottleneck principle to extract robust representations from multi-turn dialogue contexts. Two different views are constructed for each input sample and the superfluous information not related to intent detection is removed using a multi-view information bottleneck loss. Moreover, we also explore utilizing unlabeled data in Caro. A two-stage training process is introduced to mine OOD samples from these unlabeled data, and these OOD samples are used to train the resulting model with a bootstrapping approach. Comprehensive experiments demonstrate that Caro establishes state-of-the-art performances on multi-turn OOD detection tasks by improving the F1-OOD score of over 29% compared to the previous best method.

## 1 Introduction

Intent detection is vital for dialogue systems (Chen et al., 2017). Recently, promising results have been reported for intent detection under the *closed-world assumption* (Shu et al., 2017), i.e., the training and testing distributions are assumed to be identical, and all testing intents are seen in the training process. However, this assumption may not be valid in practice (Dieterich, 2017), where a deployed system usually confronts an *open-world* (Fei and Liu, 2016; Scheirer et al., 2012), i.e., the testing distribution is subject to change and Out-of-Domain (OOD) intents that are not seen in the training process may emerge in testing. It is necessary to equip intent detection modules with OOD detection abilities to accurately classify seen In-Domain (IND)

intents while rejecting unseen OOD intents (Yan et al., 2020a).

Various methods are proposed to tackle the issue of OOD detection on classification problems (Geng et al., 2020). Existing approaches include using thresholds (Zhou et al., 2021) or $(k + 1)$-way classifiers ($k$ is the number of IND classes) (Zhan et al., 2021). Promising results are reported to apply these OOD detection methods on intent detection modules (Zhou et al., 2022). However, most existing OOD intent detection studies only focus on single-turn inputs (Yan et al., 2020a; Lee and Shalyminov, 2019), i.e., only the most recently issued utterance is taken as the input. In real applications, completing a task usually necessitates multiple turns of conversations (Weld et al., 2021). Therefore, it is important to explicitly model multi-turn contexts when building OOD intent detection modules since users' intents generally depend on turns of conversations (Qin et al., 2021).

However, it is non-trivial to directly extend previous methods to the multi-turn setting (Ghosal et al., 2021). Specifically, we usually experience long distance obstacles when modeling multi-turn dialogue contexts, i.e., some dialogues have extremely long histories filled with irrelevant noises for intent detection (Liu et al., 2021). It is challenging to directly apply previous OOD intent detection methods under this obstacle since the learned representations may contain superfluous information that is irrelevant for intent detection tasks (Federici et al., 2019).

Another challenge for OOD detection in multi-turn settings is the absence of OOD samples in the training phase (Zeng et al., 2021a). Specifically, it is hard to refine learned representations for OOD detection without seeing any OOD training samples (Shen et al., 2021), and it is expensive to construct OOD samples before training, especially when multi-turn contexts are considered (Chen and Yu, 2021). Fortunately, unlabeled data (i.e., a mix-

ture of IND and OOD samples) provide a convenient way to access OOD samples since these unlabeled data are almost "free" to collect from a deployed system. However, few studies have explored utilizing unlabeled data for OOD detection in the multi-turn setting.

In this study, we propose a novel context-aware OOD intent detection framework **Caro** to address the above challenges for OOD intent detection in multi-turn settings. Specifically, we follow the information bottleneck principle (Tishby et al., 2000) to tackle the long-distance obstacle exhibited in multi-turn contexts. Robust representations are extracted by retaining predictive information while discarding superfluous information unrelated to intent detection. This objective is achieved by optimizing an unsupervised multi-view information bottleneck loss, during which two views are built based on the global pooling approach and adaptive reception fields. A gating mechanism is introduced to adaptively aggregate these two views to obtain an assembled representation. Caro also introduces a two-stage self-training scheme to mine OOD samples from unlabeled data. Specifically, the first stage builds a preliminary OOD detector with OOD samples synthesized from IND data. The second stage uses this detector to select OOD samples from the unlabeled data and use these samples to further refine the OOD detector. We list our key contributions:

1. We propose a novel framework Caro to address a challenging yet under-explored problem of OOD intent detection considering multi-turn dialogue contexts.

2. Caro learns robust representations by building diverse views of inputs and optimizing an unsupervised multi-view loss following the information bottleneck principle. Moreover, Caro mines OOD samples from unlabeled data to further refine the OOD detector.

3. We extensively evaluate Caro on multi-turn dialogue datasets. Caro obtains state-of-the-art results, outperforming the best baseline by a large margin (29.6% in the F1-OOD score).

## 2 Related Work

**OOD Detection** is a widely investigated machine learning problem (Geng et al., 2020). Recent approaches try to improve the OOD detection performance by learning more robust representations on IND data (Zhou et al., 2021; Yan et al., 2020b; Zeng et al., 2021a; Zhou et al., 2022; Wu et al., 2022) and use these representations to develop density-based or distance-based OOD detectors (Lee et al., 2018; Tan et al., 2019; Liu et al., 2020; Podolskiy et al., 2021). Some works also try to build OOD detectors with generated pseudo OOD samples (Hendrycks et al., 2018; Shu et al., 2021; Zhan et al., 2021; Marek et al., 2021) or thresholds based approaches (Gal and Ghahramani, 2016; Lakshminarayanan et al., 2017; Ren et al., 2019; Gangal et al., 2020; Ryu et al., 2017).

Some OOD detection methods also make use of unlabeled data. Existing approaches either focus on utilizing unlabeled IND data (Xu et al., 2021; Jin et al., 2022) or adopting a self-supervised learning framework to handle mixtures of IND and OOD samples (Zeng et al., 2021b). These approaches do not explicitly model multi-turn contexts.

**Modeling Multi-turn Dialogue Contexts** is the foundation for various dialogue tasks (Li et al., 2020; Ghosal et al., 2021; Chen et al., 2021). However, few works focus on detecting OOD intents in the multi-turn setting. Lee and Shalyminov (2019) proposed to use counterfeit OOD turns extracted from multi-turn contexts to train the OOD detector, and Chen and Yu (2021) augmented seed OOD samples that span multiple turns to improve the OOD detection performance. Nevertheless, these approaches either suffer from the long distance obstacle or require expensive annotated OOD samples. In this study, we attempt to learn robust representation by explicitly identifying and discarding superfluous information.

**Representation Learning** is also related to our work. Recent approaches for representation learning include optimizing a contrastive loss (Caron et al., 2020; Gao et al., 2021) or maximizing the mutual information between features and input samples (Poole et al., 2019). However, these approaches cannot tackle the long distance obstacle exhibited in multi-turn contexts. In this study, we follow the information bottleneck principle (Tishby et al., 2000; Federici et al., 2019) to remove superfluous information from long contexts.

## 3 Problem Setup

We start by formulating the problem: Given $k$ IND intent classes $\mathcal{I} = \{I_i\}_{i=1}^k$, we denote all samples that do not belong to these $k$ classes as the $(k+1)$-th intent $I_{k+1}$. Our training data contain a set of labeled IND samples $\mathcal{D}_I = \{\langle \boldsymbol{x}_i, y_i \rangle\}$ and a set of

unlabeled samples $\mathcal{D}_U = \{\langle \widetilde{x}_i, \widetilde{y}_i \rangle\}$, where $y_i \in \mathcal{I}$ and $\widetilde{y}_i \in \mathcal{I} \cup \{I_{k+1}\}$ is the label of input sample $x_i$ and $\widetilde{x}_i$, respectively. $\widetilde{y}_i$ labels are not observed during training. Our testing data contain a mixture of IND and OOD samples $\mathcal{D}_T = \{\langle \widetilde{x}_i, \widetilde{y}_i \rangle\}$, where $\widetilde{y}_i \in \mathcal{I} \cup \{I_{k+1}\}$. For a testing input $\widetilde{x}$, our OOD intent detector aims to classify the intent label of $\widetilde{x}$ if it belongs to an IND intent or reject $\widetilde{x}$ if it belongs to the OOD intent $I_{k+1}$. We also assume a validation set $\mathcal{D}_V$ that only contains IND samples is available. Moreover, each input sample $x$ from $\mathcal{D}_I$, $\mathcal{D}_U$, $\mathcal{D}_V$, and $\mathcal{D}_T$ consists of an utterance $u$ and a multi-turn dialogue history $h = u_1, \ldots, u_t$, $(t \geq 0)$ prior of $u$: $x = \langle h, u \rangle$. $u_i$ is the utterance issued in each dialogue turn.

## 4   Method

Care tackles the OOD intent detection problem by training a $(k+1)$-way classifier $F$ on $\mathcal{D}_I \cup \mathcal{D}_U$. Specifically, samples classified into the $(k+1)$-th intent $I_{k+1}$ are considered as OOD samples. There are mainly two challenges to be addressed in Caro: (1) How to alleviate the long distance obstacle and learn robust representations from multi-turn dialogue contexts; (2) How to effectively leverage unlabeled data for OOD intent detection. These two issues are tackled with two key ingredients in Caro (see Figure 1): 1. A multi-view information bottleneck method (Section 4.1); 2. A two-stage self-training scheme (Section 4.2).

### 4.1   Multi-View Information Bottleneck

The major challenge for learning robust representations from multi-turn dialogue contexts is the long distance obstacle, i.e., information that is irrelevant for intent detection may degenerate the extracted representation if the dialogue history $h$ becomes too long. In this study, we follow the information bottleneck principle (Tishby et al., 2000) to alleviate this issue, i.e., only the task-relevant information is retained in the extracted representations while all the superficial information is discarded. Specifically, we adopt a more general *unsupervised* multi-view setting for the information bottleneck method (Federici et al., 2019). For each input sample $x_i$, two semantic invariant views are constructed: $v_1(x_i), v_2(x_i)$. These two views preserve the same task-relevant information (Zhao et al., 2017). The mutual information between $v_1(x_i)$ and $v_2(x_i)$ are maximized while the information not shared between $v_1(x_i)$ and $v_2(x_i)$ are eliminated.

To achieve this goal, we adopt the multi-view information bottleneck loss introduced by Federici et al. (2019).

**Constructing Multiple Views**   for an input sample $x$ is the key to the success of the unsupervised information bottleneck method. In this study, we construct these two views $v_1(x_i), v_2(x_i)$ by adjusting the receptive fields of the final representation. This scheme is inspired by the observation in the neuroscience community that human brains process information with multiple receptive fields (Sceniak et al., 1999), i.e., the receptive field size for neurons is adapted based on input stimuli (Spillmann et al., 2015) so that different regions of inputs are emphasized (Pettet and Gilbert, 1992). This phenomenon has been demonstrated to be effective in modeling more robust features (Pandey et al., 2022) and inspired numerous successful neural models (Wang et al., 2021; Wei et al., 2017).

Specifically, for each input sample $x = \langle h, u \rangle$, we first concatenate all utterances in $x$ and then use a pre-trained BERT model $E$ (Devlin et al., 2018) to encode the sequence of concatenated tokens into a sequence of embedding vectors $E(x) = [e_1, \cdots e_n]$, where $e_i \in \mathbb{R}^m$. The following two strategies are used to construct two different views:

1. *Global Pooling* builds view $v_1(x)$ with a mean-pooling layer on top of $[e_1, \cdots e_n]$, $v_1(x)$ assumes each token embedding is equally weighted:

$$v_1(x) = \sum_{i=1}^{n} e_i / n \qquad (1)$$

2. *Adaptive Reception Field* builds view $v_2(x)$ by adapting the synaptic weight of each token embedding based on the input $x$:

$$v_2(x) = \sum_{i=1}^{n} \frac{\exp(\alpha_i)}{\sum_{j=1}^{n} \exp(\alpha_j)} \cdot e_i \qquad (2)$$
$$\alpha_i = \sigma(w_i \cdot \text{ReLU}(W_1 \cdot s)),$$

where $s \in \mathbb{R}^{nm}$ is the concatenation of all $n$ embeddings $[e_1, \cdots e_n]$. $\sigma$ is the Sigmoid activation function. $w_i \in \mathbb{R}^{1 \times r_1}$ $(i = 1, \cdots n)$ and $W_1 \in \mathbb{R}^{r_1 \times nm}$ are learnable parameters. $r_1$ is the size of the intermediate layer. Moreover, to enhance the generalization ability, we set a small value for $r_1$ in our implementation to form a bottleneck structure in the weighting function (Hu et al., 2017).

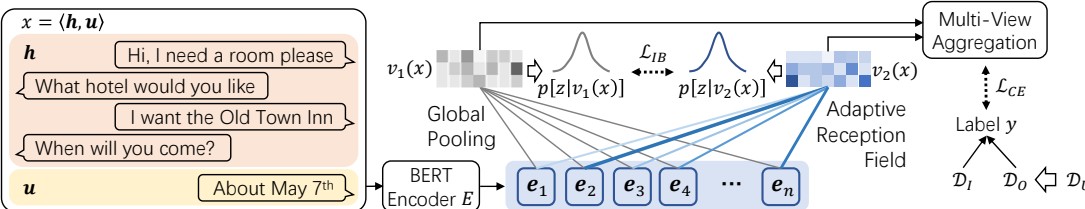

Figure 1: Framework of Caro. For each input sample $x = \langle h, u \rangle$, two views $v_1(x)$ and $v_2(x)$ are obtained and a multi-view information bottleneck loss $\mathcal{L}_{IB}$ is optimized to learn robust representations. A two-stage training process is introduced to mine OOD samples $\mathcal{D}_O$ from unlabeled data $\mathcal{D}_U$, and optimize the cross entropy loss $\mathcal{L}_{CE}$ with $\mathcal{D}_O \cup \mathcal{D}_I$

**Optimizing Information Bottleneck** is performed in an unsupervised setting based on the two views of each sample. Specifically, we assume the representation $z_i$ of each view $v_i(x)$, $(i = 1, 2)$ follows a distribution that is parameterized by an encoder $p(z|v_i)$, where $v_i$ is short for $v_i(x)$ for abbreviation. To facilitate the computation, we model $p$ as factorized Gaussian distributions, i.e., $p(z|v_i) = \mathcal{N}[\mu(v_i), \Sigma(v_i)]$, in which $\mu(v_i)$ and $\Sigma(v_i)$ are two neural networks that produce the mean and deviation, respectively. The following information bottleneck loss (Federici et al., 2019) is optimized to remove superfluous information in $v_1(x)$ and $v_2(x)$:

$$\mathcal{L}_{IB} = -I(z_1; z_2) + \frac{1}{2}(D_{KL}[p(z|v_1)||p(z|v_2)] + D_{KL}[p(z|v_2)||p(z|v_1)]), \quad (3)$$

where $I$ calculates the mutual information of two random variables, and $D_{KL}$ calculates the KL divergence between two distributions.

### 4.2 Two-stage Self-training

Although robust representations can be obtained with the help of the information bottleneck loss $\mathcal{L}_{IB}$ from Section 4.1, we still lack the annotations for OOD samples to train the $(k+1)$-way classifier $F$ for OOD detection. In this study, we tackle this issue with a two-stage self-training process, which mines OOD samples from the unlabeled data $\mathcal{D}_U$ with a bootstrapping approach. Moreover, for each input sample $x$, we also aggregate its two views $v_1(x)$ and $v_2(x)$ with a dynamic gate to obtain assembled representations in training.

**Stage One** synthesizes pseudo OOD samples $\mathcal{D}_P$ by mixing up IND features. Specifically, samples from $\mathcal{D}_I$ are first mapped into IND representation vectors, and pseudo OOD samples are obtained as convex combinations of these vectors (Zhan et al., 2021). A preliminary OOD detector $F$ is trained

using the classical cross-entropy loss $\mathcal{L}_{CE}$ on these synthesized pseudo OOD samples and labeled IND samples $\mathcal{D}_I$. This stage endows $F$ with a preliminary ability to predict the intent distribution of each input sample.

**Stage Two** predicts a pseudo label for each sample $x \in \mathcal{D}_U$ using $F$, and then collects samples that are assigned with the OOD label $I_{k+1}$ as a set of mined OOD samples $\mathcal{D}_O$. With the help of $\mathcal{D}_O$, we further train the classifier $F$ on the following loss:

$$\mathcal{L} = \mathop{\mathbb{E}}_{x \in \mathcal{D}_I \cup \mathcal{D}_O} \mathcal{L}_{CE} + \lambda \mathop{\mathbb{E}}_{x \in \mathcal{D}_U} \mathcal{L}_{IB} \quad (4)$$

where $\lambda$ is a scalar hyper-parameter to control the weight of the information bottleneck loss.

**Multi-view Aggregation** is performed to obtain assembled representations for input samples. Specifically, whenever we need to extract the representation $v(x)$ for an input sample $x$ in the training process, we use the following aggregation approach:

$$\begin{aligned} v(x) &= \beta \otimes v_1(x) + (1 - \beta) \otimes v_2(x) \\ \beta &= \sigma(W_3 \cdot \text{ReLU}(W_2 \cdot (v_1(x) + v_2(x)))) \end{aligned} \quad (5)$$

where $\otimes$ represents the element-wise product, $W_2 \in \mathbb{R}^{r_2 \times m}$ and $W_3 \in \mathbb{R}^{m \times r_2}$ are learnable parameters. $r_2$ is the size of the intermediate layer.

The training of Caro is given in Algorithm 1.

## 5 Experiments

### 5.1 Datasets

We perform experiments on two variants of the STAR dataset (Mosig et al., 2020), i.e., STAR-Full and STAR-Small. Specifically, STAR is a task-oriented dialogue dataset that has 150 intents. It is designed to model long context dependence, and provides explicit annotations of OOD intents. Following Chen and Yu (2021), we regard samples

**Algorithm 1:** The training process of Caro

**Input:** IND data $\mathcal{D}_I$, unlabeled data $\mathcal{D}_U$.

**Output:** A trained OOD detector $F$.

```
// Stage 1
```

1 Synthesize pseudo OOD samples $\mathcal{D}_P$ by mixing up IND representations.

2 Train $F$ using the cross-entropy loss $\mathcal{L}_{CE}$ on $\mathcal{D}_I \cup \mathcal{D}_P$.

```
// Stage 2
```

3 Mine OOD samples $\mathcal{D}_O$ from $\mathcal{D}_U$ using $F$.

4 Train $F$ using $\mathcal{L}$ (Eq. 4) on $\mathcal{D}_I$, $\mathcal{D}_O$, and $\mathcal{D}_U$

| | Train | | Valid $\mathcal{D}_V$ | Test $\mathcal{D}_T$ | # Avg. Context Turns |
|---|---|---|---|---|---|
| | $\mathcal{D}_I$ | $\mathcal{D}_U$ | | | |
| STAR-Full | 15.4K | 7.9K | 2.8K | 2.9K | 6.13 |
| STAR-Small | 7.7K | 3.9K | 2.8K | 2.9K | 6.12 |

Table 1: Dataset statistics.

from intents "out_of_scope", "custom", or "ambiguous" as OOD samples and all other samples as IND samples. We also filter out generic utterances (e.g., greetings) in the pre-processing stage.

STAR-Full contains all pre-processed samples from the original STAR dataset. To construct unlabeled data $\mathcal{D}_U$, we extract 30% of IND samples and all OOD samples from the training set. The intent labels of all these extracted samples are removed, and the remaining samples in the training set are used as the labeled data $\mathcal{D}_I$. STAR-Small is constructed similarly, except that we down-sample 50% of the training set. We aim to evaluate the performance of OOD detection in low-resource scenarios with STAR-Small. Table 1 shows the statistics of these datasets.

### 5.2 Metrics

Following Zhang et al. (2021b); Shu et al. (2021), the OOD intent detection performance of our model is evaluated using the macro F1-score (**F1-All**) over all testing samples (i.e., IND and OOD samples). The fine-grained performance of our model is also evaluated by the macro F1-score over all IND samples (**F1-IND**) and OOD samples (**F1-OOD**), respectively. We use macro F1-scores to handle the class imbalance issue of the test set.

### 5.3 Implementation Details

Our BERT backbone is initialized with the pretrained weights of BERT-based-uncased (Devlin et al., 2018). We use AdamW, and Adam (Kingma and Ba, 2014) to fine-tune the BERT backbone and all other modules with a learning rate of 1e-5 and 1e-4, respectively. The Jensen-Shannon mutual information estimator (Hjelm et al., 2018) is used to estimate the mutual information $I$ in Eq. 3. All results reported in our paper are averages of 3 runs with different random seeds. Hyper-parameters are searched based on IND intent classification performances on the validation set. See Appendix A for more implementation details. Note that Caro only introduces little computational overhead compared to other OOD detection models (See Appendix C).

### 5.4 Baselines

Our baselines can be classified into two categories based on whether they use unlabeled data. The first set of baselines only use labeled IND samples $\mathcal{D}_I$ in training: **1. MSP**: (Hendrycks and Gimpel, 2017) utilizes the maximum Softmax predictions of a $k$-way IND classifier to detect OOD inputs. We set the OOD detection threshold to 0.5 following Zhang et al. (2021a); **2. SEG**: (Yan et al., 2020b) proposes a semantic-enhanced Gaussian mixture model; **3. DOC**: (Shu et al., 2017) employs $k$ 1-vs-rest Sigmoid classifiers and uses the maximum predictions to detect OOD intents; **4. ADB**: (Zhang et al., 2021b) learns an adaptive decision boundaries for OOD detection; **5. DAADB**: (Zhang et al., 2021c) improves the baseline ADB with distance-aware intent representations; **6. Outlier**: (Zhan et al., 2021) mixes convex interpolated outliers and open-domain outliers to train a $(k + 1)$-way classifier for OOD detection; **7. CDA**: (Lee and Shalyminov, 2019) utilizes counterfeit OOD turns to detect OOD samples.

The second set of baselines uses both labeled IND samples $\mathcal{D}_I$ and unlabeled samples $\mathcal{D}_U$ for training. Specifically, Zeng et al. (2021b) proposes a self-supervised contrastive learning framework ASS to model discriminative features from unlabeled data with an adversarial augmentation module. We implement three variants of ASS by using different detection modules: **1. ASS+MSP**: uses the detection module from the baseline MSP; **2. ASS+LOF**: (Lin and Xu, 2019) implements the OOD detector as the local outlier factor; **3. ASS+GDA**: (Xu et al., 2020a) uses a generative distance-based classifier with Mahalanobis distance as the detection module.

Moreover, we also report the performance of a

| Model | STAR-Full | | | STAR-Small | | |
|---|---|---|---|---|---|---|
| | F1-All | F1-OOD | F1-IND | F1-All | F1-OOD | F1-IND |
| Oracle | 50.1 | 64.46 | 50 | 46.54 | 58.23 | 46.46 |

| | Model | F1-All | F1-OOD | F1-IND | F1-All | F1-OOD | F1-IND |
|---|---|---|---|---|---|---|---|
| $\mathcal{D}_I$ | MSP | 40.83 | 19.74 | 40.97 | 37.17 | 18.1 | 37.31 |
| | MSP w/o $h$ | 17.29 | 14.12 | 17.31 | 17.12 | 13.49 | 17.14 |
| | SEG | 17.45 | 6.85 | 17.53 | 11.66 | 7.39 | 11.69 |
| | SEG w/o $h$ | 0.06 | 2.77 | 0.04 | 0.05 | 2.27 | 0.04 |
| | DOC | 26.53 | 16.80 | 26.60 | 3.47 | 11.78 | 3.41 |
| | DOC w/o $h$ | 11.31 | 14.16 | 11.29 | 0.08 | 11.04 | 0 |
| | ADB | 44.64 | 20.56 | 44.80 | 41.36 | 18.23 | 41.51 |
| | ADB w/o $h$ | 23.27 | 17.63 | 23.30 | 20.08 | 21.27 | 20.07 |
| | DAADB | 37.27 | 22.87 | 37.37 | 34.81 | 20.43 | 34.91 |
| | DAADB w/o $h$ | 17.87 | 15.15 | 17.88 | 16.34 | 17.03 | 16.33 |
| | Outlier | 43.84 | 19.53 | 44.01 | 39.51 | 19.92 | 39.64 |
| | Outlier w/o $h$ | 23.35 | 16.75 | 23.39 | 19.56 | 15.42 | 19.59 |
| | CDA | 43.76 | 5.26 | 44.03 | 40.02 | 10.48 | 40.22 |
| $\mathcal{D}_I+\mathcal{D}_U$ | ASS+MSP | 41.97 | 25.15 | 42.08 | 40.85 | 19.47 | 40.99 |
| | ASS+LOF | 39.87 | 17.65 | 40.02 | 39.54 | 18.49 | 39.68 |
| | ASS+GDA | 43.73 | 21.24 | 43.88 | 40.86 | 16.72 | 41.02 |
| | **Caro** (ours) | **48.75**($\pm$1.0) | **54.75**($\pm$3.2) | **48.71**($\pm$1.0) | **45.02**($\pm$1.1) | **46.78**($\pm$1.8) | **45.01**($\pm$1.1) |

Table 2: Performance of Caro and baselines. All results are averages of three runs and the best results are bolded. The standard deviation of the performance of Caro is provided in parentheses.

$(k + 1)$-way classifier trained on fully labeled IND and OOD samples (**Oracle**), i.e., we preserve all labels for samples in $\mathcal{D}_I$ and $\mathcal{D}_U$. This model is generally regarded as the upper bound of our model since it uses all the annotations.

For fair comparisons, all baselines use the same pretrained BERT-base backbones as our model. Multi-turn dialogue contexts in all baselines are modeled by concatenating utterances in dialogue histories. Moreover, to further validate the importance of dialogue contexts for OOD detection, we also implement a single-turn variant for the first set of baselines by ignoring multi-turn dialogue contexts (**w/o $h$**), i.e., only the latest user issued utterance $u$ is used as the input. Note that we do not implement the single-turn variant for the baseline CDA since CDA is specifically designed to utilize multi-turn contexts. See Appendix B for more details about baselines.

### 5.5 Main Results

The results for our model Caro and all baselines are shown in Table 2. It can be seen that Caro outperforms all other baselines on both datasets with large margins. We highlight several observations: **1.** Methods that model multi-turns of dialogue histories (e.g., MSP, SEG, DOC, ADB, DA-ADB, and Outlier) generally outperform their single turn counter (i.e., models marked with "w/o $h$") with large margins. This validates our claim that it is necessary to consider multi-turn dialogue contexts

for OOD intent detection since users' intents may depend on prior turns. **2.** Our method Caro outperforms all baselines that only use IND data $\mathcal{D}_I$. The performance gain demonstrates the advantage of incorporating unlabeled data for OOD detection, which can be used to learn compact representations for both IND and OOD intents. **3.** Caro also outperforms baselines that utilize unlabeled data $\mathcal{D}_U$. This validates Caro's effectiveness in tackling the long distance obstacle and modeling unlabeled samples. Our baselines are prone to capture irrelevant noises for OOD intent detection, while Caro incorporates multi-view information bottleneck loss to remove superfluous information.

We also analyze the effect of unlabeled data size (Appendix E) and $\lambda$ (Appendix F) on the OOD intent detection performance and carry out a case study (Appendix G).

### 5.6 Ablation Studies

To validate our motivation and model design, we ablate our model components and loss terms.

**Model Components:** Ablation studies are carried out to validate the effectiveness of each component in Caro. Specifically, the following variants are investigated: **1. w/o $\mathcal{D}_U$** removes training stage two, i.e., only $\mathcal{D}_I$ is used for training. **2. w/o MV** ablates the multi-view construction approach introduced in Caro. Specifically, we adopt the approach used by Gao et al. (2021) to perform two dropouts

| Model | STAR-Full | | | STAR-Small | | |
|---|---|---|---|---|---|---|
| | F1-All | F1-OOD | F1-IND | F1-All | F1-OOD | F1-IND |
| Caro | **48.75** | **54.75** | **48.71** | **45.02** | **46.78** | **45.01** |
| w/o $\mathcal{D}_U$ | 45.97 | 21.45 | 46.14 | 42.24 | 23.23 | 42.37 |
| w/o MV | 47.71 | 53.35 | 47.67 | 44.42 | 38.89 | 44.46 |
| w/o VA | 47.34 | 50.85 | 47.32 | 44.14 | 43.88 | 44.15 |
| w/o IB | 48.23 | 49.37 | 48.22 | 44.14 | 37.06 | 44.19 |

Table 3: Ablation on different components of Caro.

| Model | STAR-Full | | | STAR-Small | | |
|---|---|---|---|---|---|---|
| | F1-All | F1-OOD | F1-IND | F1-All | F1-OOD | F1-IND |
| Caro | **48.75** | **54.75** | **48.71** | **45.02** | **46.78** | **45.01** |
| InfoMax | 47.27 | 49.92 | 47.25 | 44.27 | 36.66 | 44.32 |
| MVI | 48.46 | 51.99 | 48.44 | 44.70 | 36.16 | 44.76 |
| CL | 48.18 | 52.54 | 48.15 | 44.59 | 35.31 | 44.65 |
| SimCSE | 47.73 | 47.74 | 47.73 | 44.30 | 27.02 | 44.42 |

Table 4: Ablation on the representation learning loss.

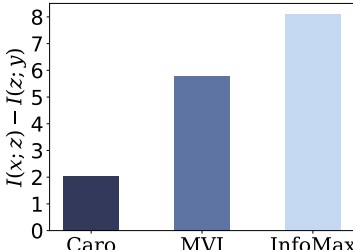

Figure 2: Comparing representations obtained by different objectives on the STAR-Full dataset. A lower score means that the learned representation discards more superficial information. See Appendix D for measurements used to produce the graph.

with two different masks when constructing these two views. **3. w/o VA** ablates the multi-view aggregation approach, i.e., the representations of two views are directly added instead of using the adaptive gate in Eq. 5. **4. w/o IB** removes the information bottleneck loss $\mathcal{L}_{IB}$. We implement this variant by setting $\lambda = 0$ in Eq.4.

Results in Table 3 indicate that Caro outperforms all ablation variants. Specifically, we can also observe that: 1. Training models without unlabeled data (i.e., w/o $\mathcal{D}_U$) degenerate the performance of Caro by a large margin. The F1-OOD score suffers an absolute decrease of 33.3% and 23.6% on STAR-FULL and STAR-Small, respectively. This validates our claim that effective utilization of unlabeled data improves the performance of OOD detection. 2. Our multi-view construction approach helps to improve the OOD detection performance (see w/o MV), and our multi-view aggregation approach also benefits the extracted representation (see w/o VA). 3. Removing the multi-view information bottleneck loss (i.e., w/o IB) degenerates the OOD performance. This validates our claim that multi-turn contexts may contain irrelevant noises for OOD intent detection.

**Information Bottleneck Loss:** We further demonstrate the effectiveness of our information bottleneck loss $\mathcal{L}_{IB}$ by replacing $\mathcal{L}_{IB}$ in Eq. 4 with other alternatives of representation learning. Specifically, assume $x$ is an input sample. **1. InfoMax** (Poole et al., 2019) maximizes the mutual information between $x$ and its representation $z$: $I(x; z)$; **2. MVI** (Bachman et al., 2019) is similar

to InfoMax except that it maximizes the mutual information between $x$'s two views $I(v_1(x); v_2(x))$; Note that both InfoMax and MVI do not attempt to remove superficial information from representations. **3. CL** (Caron et al., 2020) uses a contrastive learning loss. Positive pairs in this variant are obtained using our multi-view construction approach. **4. SimCSE** (Gao et al., 2021) is similar to CL except that it acquires positive pairs by two different dropouts on the BERT encoder.

Results in Table 4 show that the information bottleneck loss used in Caro performs better than all other variants. We also want to highlight that the approach of explicitly removing superficial information in Caro makes it outperform InfoMax and MVI by 4.83% and 2.76%, respectively, on the F1-OOD score. This validates our claim that long contexts may contain superficial information that degenerates intent detection, and the multi-view information bottleneck loss used in Caro effectively removes this superficial information.

Moreover, we also perform fine-grained analysis of the learned representations following Tishby et al. (2000). Specifically, for an input sample $x$ with a label of $y$ and an extracted representation of $z$, two scores are calculated: 1. Observational information score (measured by $I(x; z)$); 2. Predictive ability score (measured by $I(z; y)$). An ideal representation would be maximally predictive about the label while retaining a minimal amount of information from the observations (Tishby et al., 2000; Federici et al., 2019). Here we report the score of $I(x; z) - I(z; y)$ for Caro, MVI and InforMax in Figure 2. It can be seen that the information bottleneck loss helps Caro to achieve the lowest $I(x; z) - I(z; y)$ score. This indicates that representations learned in Caro retrain low observational information while achieving a relatively

| Context Len | | F1-All | F1-OOD | F1-IND |
|---|---|---|---|---|
| Long | w/o IB | 44.14 | 37.06 | 44.19 |
| | w IB | 45.02 (+0.88) | 46.78 (+9.72) | 45.01 (+0.82) |
| Short | w/o IB | 43.61 | 40.68 | 43.63 |
| | w IB | 43.70 (+0.09) | 43.32 (+2.64) | 43.70 (+0.07) |

Table 5: Benefit of $\mathcal{L}_{IB}$ under different context lengths on the STAR-Small dataset. Long context means retaining all the original dialogue contexts (6 turns on average), and short context means truncating contexts longer than 3 turns. Scores in parentheses is the performance improvement brought by $\mathcal{L}_{IB}$

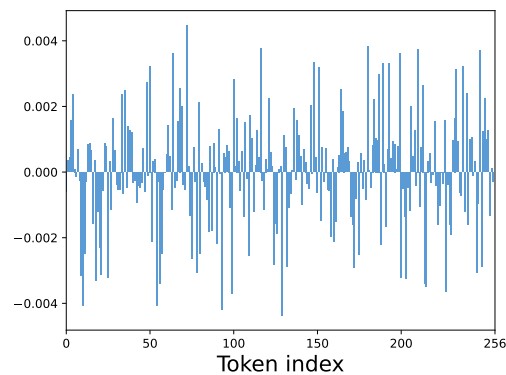

Figure 3: Difference of averaged weight score at each token index for testing samples from STAR-Full.

high predictive ability.

### 5.7 Further Analysis

**Benefit of $\mathcal{L}_{IB}$ in Different Context Lengths**
We also validate the benefit of our information bottleneck loss $\mathcal{L}_{IB}$ (Eq. 3) under different context lengths. Specifically, we construct a variant of STAR-Small (denoted as "Short") by truncating contexts longer than 3 turns, i.e., the dialogue histories before the latest 3 turns are discarded. We also denote the original STAR-Small dataset as "Long", which has a maximum context length of 7 turns. Caro's performance with and without $\mathcal{L}_{IB}$, i.e., "w IB" and "w/o IB" is tested on these two datasets.

Results in Table 5 show that Caro benefits more from $\mathcal{L}_{IB}$ in longer contexts. Specifically, the longer the context, the larger improvement is brought by $\mathcal{L}_{IB}$ on the OOD detection performance. This further validates our claim that our information bottleneck loss $\mathcal{L}_{IB}$ helps remove superficial information unrelated to intent detection.

**Diversity of Adaptive Reception Field** Our multi-view information bottleneck objective expects two diverse views for each input sample (Federici et al., 2019). Here we validate the

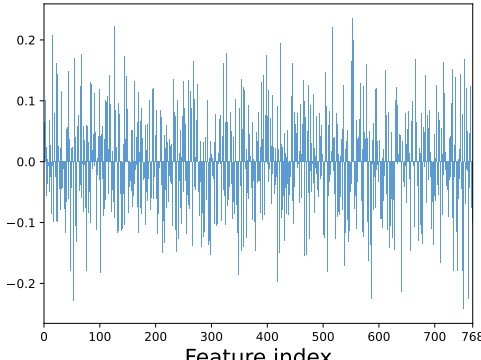

Figure 4: Difference of averaged aggregation weights at each dimension for testing samples in STAR-Full.

diversity of our two views (Section 4.1) by visualizing the distribution of weight score $\alpha_i$ in Eq. 2. Specifically, we first calculate the average weight scores received at each token index for samples from the same intent (we use a max sequence length of 256). Then we choose two intents (i.e., *weather_inform_forecast* and *trip_inform_simple_step_ask_proceed*) and visualize the difference between their averaged weight score at each token index in Figure 3. It can be seen that weight scores change sharply across different intents and token indices. That means the view $v_2(\boldsymbol{x})$ constructed for each sample is diverse.

**Analysis of Aggregation Weights** We also visualize the weight $\beta$ used in the multi-view aggregation process (Eq. 5). Specifically, we expect these two views in Eq. 5 to receive different weights. Concretely, we first calculate the averaged $\beta$ vector for all testing samples from STAR-Small. Then we calculate the difference of weights received by these two views $v_1(\boldsymbol{x})$ and $v_2(\boldsymbol{x})$ in Eq. 5, and visualize values in each dimension in Figure 4. It can be seen that diverse weights are used in the multi-view aggregation process.

## 6 Conclusion

In this paper, we propose Caro, a novel OOD intent detection framework to explore OOD detection in multi-turn settings. Caro learns robust representations by building diverse views of an input and optimise an unsupervised multi-view loss following the information bottleneck principle. OOD samples are mined from unlabeled data, which are used to train a $(k+1)$-way multi-view classifier as the resulting OOD detector. Extensive experiments demonstrate that Caro is effective as modeling multi-turn contexts and outperforms SOTA baselines.

## Limitations

One major limitation of this work is its input modality. Specifically, our method is limited to textual inputs and ignores inputs in other modalities such as audio, vision, or robotic features. These modalities provide valuable information that can be used to build better OOD detectors. In future works, we will try to model multi-modal multi-turn contexts for OOD intent detection.

## Ethics Statement

This work does not present any direct ethical issues. In the proposed work, we seek to develop a context-aware method for OOD intent detection, and we believe this study leads to intellectual merits that benefit from a reliable application of NLU models. All experiments are conducted on open datasets.

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

## A    More Implementation Details

We use Huggingface's Transformers library (Wolf et al., 2020) and train with the backbone of BERT (Devlin et al., 2018). The max_seq_length is 256 for BertTokenize. The classification head is implemented as two-layer MLPs with the LeakyReLU activation (Xu et al., 2020b), while the projection heads in $\mu(v_i)$ and $\Sigma(v_i)$ as three-layer MLPs. The projection dimension is 64. Following (Zhan et al., 2021), We use AdamW (Kingma and Ba, 2014) to fine-tune BERT using a learning rate of 1e-5 and Adam (Wolf et al., 2019) to train the MLP heads using a learning rate of 1e-4. Following (Federici et al., 2019), we use Jensen-Shannon mutual information estimator (Hjelm et al., 2018) to maximize mutual information between two random variables. In the training stage, 15 epochs of pre-training are first conducted, and then 10 epochs of training are conducted by adding the process of unsupervised representation learning on unlabeled data with early stopping. The batch size is 25 for IND and unlabeled datasets, respectively. We set the weight $\lambda$ for $\mathcal{L}_{IB}$ to be 0.5 in all experiments. And we set $r_1 = 16$ and $r_2 = 48$. All results reported in our paper are averages of 3 runs with different random seeds, and each run is stopped when we reach a plateau on the validation performance. Hyper-parameters are searched based on IND intent classification performances on the validation set. All experiments are conducted in the Nvidia Tesla V100-SXM2 GPU with 32G graphical memory.

## B    More Details about Baselines

We get the baseline results (MSP, SEG, DOC, ADB, and DA-ADB) using the OOD detection toolkit TEXTOIR (Zhang et al., 2021a). We get the baseline result of Outlier by running their released codes (Zhan et al., 2021). We re-implement CDA by using counterfeit OOD turns (Lee and Shalyminov, 2019). We re-implement ASS (Zeng et al., 2021b) based on the code of authors (Zeng et al., 2021a). For fair comparisons, all baselines are implemented by using BERT as the backbone.

## C    Computational Cost Analysis

| Methods | #Para. | Training Time | Testing Time |
|---------|--------|---------------|--------------|
| Outlier | 111.47 M | 7.26 min | 14.46 s |
| Caro | 116.80 M | 8.75 min | 14.53 s |

Table 6: Number of parameters (Million), average training time for each epoch (minutes) and the total time for testing (seconds) on STAR-Full dataset.

We compare the computational cost of a vanilla OOD detector Outlier (Zhan et al., 2021) and Caro. We use the STAR-Full dataset for this analysis. As shown in Table 6, Caro only introduces marginal parameter overhead. We can also observe that using Caro only introduces a little time overhead compared to Outlier.

## D    More Details about Measurements Used to Produce the Graph

The mutual information estimation ($I(\boldsymbol{x}; \boldsymbol{z})$ and $I(\boldsymbol{z}; \boldsymbol{y})$) reported in Figure 2 are computed by training two estimation networks from scratch on the final representation of Caro. Following (Federici et al., 2019), we use Jensen-Shannon mutual information estimator (Hjelm et al., 2018) to maximize mutual information between two random variables.

The two estimation architectures consist of three-layer MLPs. We report average numerical estimations of mutual information using an energy-based bound (Poole et al., 2019) on the test dataset. To reduce the variance of the estimator, the lowest and highest $5\%$ are removed before averaging.

## E Analysis for Unlabeled Data Size

Table 7 demonstrates the effect of unlabeled data size for Caro. We downsample 100%, 75%, 50%, and 25% of the unlabeled data from STAR-Small and evaluate the performance of Caro. It can be seen that our method Caro achieves superior OOD detection performance in term of F1-OOD along with the increase of unlabeled data.

| DownSample-Rate | F1-All | F1-OOD | F1-IND |
|:---:|:---:|:---:|:---:|
| 100% | 45.02 | 46.78 | 45.01 |
| 75% | 44.40 | 37.77 | 44.44 |
| 50% | 45.04 | 30.13 | 45.15 |
| 25% | 44.47 | 20.76 | 44.62 |

Table 7: Effect of unlabeled data size on the OOD intent detection performance. The reported performance are produced on the STAR-Small dataset.

## F Analysis for Loss Weight $\lambda$

Tabel 8 reports the OOD detection results as we vary the weight $\lambda$ for $\mathcal{L}_{IB}$ in Eq. 4. The results indicate that a relatively small weight is desirable.

| $\lambda$ | F1-All | F1-OOD | F1-IND |
|:---:|:---:|:---:|:---:|
| 0.3 | 47.57 | 55.68 | 47.51 |
| 0.4 | 48.10 | 51.44 | 48.08 |
| 0.5 | **48.75** | **54.75** | **48.71** |
| 0.6 | 47.15 | 50.84 | 47.12 |
| 0.7 | 47.83 | 50.60 | 47.81 |

Table 8: Effect of $\lambda$ on the OOD intent detection performance. The reported performance are produced on the STAR-Full dataset.

## G Case Study

| Dialogue | | Prediction of "Outlier" | Prediction of "Caro" |
| --- | --- | --- | --- |
| Customer | Agent | | |
| Somebody has transferred $500 from my account. | | bank_ask_pin | OOD |
| | Would you like to file a fraud report? | | |
| yes please | | | |
| | Could I get your full name, please? | | |
| Jane Doe | | | |
| | Can you tell me your account number, please? | | |
| What am I doing on this chat? | | | |
| Hey there! Can you tell me what the weather will be tomorrow please? | | weather _inform _forecast | OOD |
| | It will be Raining all day on Saturday in Los Angeles, with temperatures of around 19 degrees celsius. | | |
| oh dang when's the next time it will be sunny? | | | |
| | It will be Sunny all day on Friday in Los Angeles, with temperatures of around 11 degrees celsius. | | |
| Nice! I think I'll head to La brea tar pits. I've never been there. have you seen those animals? Crazy | | | |
| Schedule a viewing at Shadyside apartment for Thursday at 8pm my name is john | | bank_inform _cannot _authenticate | OOD |
| | Have you already paid the application fee for the apartment? | | |
| I can't remember | | | |
| | I am sorry, but there is no viewing available at your preferred time. | | |
| I have a hot date by 8pm o Thursday | | | |
| | When would you like the viewing to start? | | |
| How is the apartment like? | | | |

Table 9: Case study of classified intents on the OOD samples (from STAR-Full dataset) by Outlier and Caro. OOD samples are classified as one of the IND classes by Outlier, which are detected as the OOD intent by Caro.

