# OpenReview forum: "Out-of-Domain Intent Detection Considering Multi-turn Dialogue Contexts"
_EMNLP/2023/Conference — Submitted to EMNLP 2023_

### Official Review · Reviewer_H5LW · 2023-08-01

**Soundness:** 2

**Excitement:**

3: Ambivalent: It has merits (e.g., it reports state-of-the-art results, the idea is nice), but there are key weaknesses (e.g., it describes incremental work), and it can significantly benefit from another round of revision. However, I won't object to accepting it if my co-reviewers champion it.

**Paper Topic And Main Contributions:**

This paper proposes context aware OOD intent detection (caro) framework to model multi-turn contexts in OOD intent detection tasks. Moreover, Caro introduces a two-stage self-training scheme to mine OOD samples from unlabeled data. The main contribution of this paper is to show the better performance of Caro compared with other models on variants of STAR dataset.

**Questions For The Authors:**

Question 1:
This paper try to solve OOD intent detection in the case where  "the testing distribution is subject to change and out-of-domain (OOD) intentions that are not seen in the training process may emerge in testing". How is the situation in which the testing distribution has changed replicated in the evaluation data created manually?


Question 2:
In Section 3.2, it seems arbitrary that 30% should be added. In order to make Du, the labels of OOD and IND are used, but since it is originally an unlabeled dataset, it seems unnatural that the ratio of IND and OOD is known. If you want to combine IND and OOD to make an unlabeled dataset, I think it might be better to make a few patterns with the ratio changed from 0% to 100% to see its robustness.

Question 3:
In this paper, experimental results are shown only for STAR data, but Chen and Yu (2021) also showed the results on FLOW and ROSTD. With only one data set, it is difficult to know whether the method works for a particular data set or works generally. I think it’s better to show the results on other datasets too.


Question 4:
Although it seems that there is no table showing the number of IND/OOD data, it is better to show the statistics.

**Reasons To Accept:**

We can see Caro (a context ware OOD intent detection ) works well on the new evaluation dataset created from STAR dataset.

**Reasons To Reject:**

We cannot confirm whether the issue set here has been solved with the evaluation dataset used in the experiment.

**Reproducibility:**

2: Would be hard pressed to reproduce the results. The contribution depends on data that are simply not available outside the author's institution or consortium; not enough details are provided.

**Reviewer Confidence:**

3: Pretty sure, but there's a chance I missed something. Although I have a good feel for this area in general, I did not carefully check the paper's details, e.g., the math, experimental design, or novelty.

---

> ### Author Rebuttal · Authors · 2023-08-28
>
> Thank you for your helpful comments! We are glad you noted the empirical benefits. We address your thoughts point by point below.
>
> --"We cannot confirm whether the issue set here has been solved with the evaluation dataset used in the experiment."
>
> 1. Our method is mainly evaluated in a multi-turn dialogue dataset, STAR, which is designed to model long context dependence with explicit OOD intent annotations. Meanwhile, we curate two variants of the STAR dataset (STAR-Full and STAR-Small) to evaluate the generalization of our method. Comprehensive experiments demonstrate that our method establishes state-of-the art performances on multi-turn OOD detection tasks by improving the F1-OOD score of over 29% compared to the previous best method.
>
> 2. Given existing OOD intent detection approaches often focus on single-turn interactions, it is really difficult for us to find additional datasets with muti-turn dialogue contexts and annotations of OOD intent.
>
> --"This paper try to solve OOD intent detection in the case where "the testing distribution is subject to change and out-of-domain (OOD) intentions that are not seen in the training process may emerge in testing". How is the situation in which the testing distribution has changed replicated in the evaluation data created manually?"
>
> 1. We regard samples from intents “out_of_scope”, “custom”, or “ambiguous” as OOD samples and all other samples as IND samples. The training dataset only contains IND samples, while the testing dataset contains both IND and OOD samples.
>
> --"In Section 3.2, it seems arbitrary that 30% should be added."
>
> 1. We formulate our problem in Section 3, where D_U is defined as a set of unlabeled samples (IND and OOD samples). Hence, we did not add specific 30% in it.
>
> --" If you want to combine IND and OOD to make an unlabeled dataset, I think it might be better to make a few patterns with the ratio changed from 0% to 100% to see its robustness."
>
> 1. We extract 30% of IND samples and all OOD samples from the orignial STAR training set to make the unlabeled dataset. The resulting unlabeled dataset contains more IND samples compared to OOD samples (6615 IND samples, 1248 OOD samples), which reflects the practical scenario that the majority of test data may remain IND [1]. Meanwhile, relatively a small number of OOD samples in the unlabeled dataset translates into a harder learning problem, because the IND training set and the unlabeled set become largely overlapping, which makes the unlabeled set suitable for evaluating the robustness of our method.
> 2. We will add more experiments with different mixing ratios in our revised paper.
> [1] Training ood detectors in their natural habitats, ICML2022.
>
> --"In this paper, experimental results are shown only for STAR data, but Chen and Yu (2021) also showed the results on FLOW and ROSTD. With only one data set, it is difficult to know whether the method works for a particular data set or works generally. I think it’s better to show the results on other datasets too."
>
> 1. Kindly note that ROSTD consists of single-turn commands while we aim to study OOD intent detection with multi-turn dialogue contexts.
> 2. FLOW is originally built for semantic parsing (Chen and Yu, 2021), and unfortunately there are no explict annotations of OOD intents.
>
> --"Although it seems that there is no table showing the number of IND/OOD data, it is better to show the statistics."
>
> 1. We will show it in our next version.

---

### Official Review · Reviewer_dLzu · 2023-08-04

**Soundness:** 3

**Ethical Concerns:**

Yes

**Excitement:**

3: Ambivalent: It has merits (e.g., it reports state-of-the-art results, the idea is nice), but there are key weaknesses (e.g., it describes incremental work), and it can significantly benefit from another round of revision. However, I won't object to accepting it if my co-reviewers champion it.

**Paper Topic And Main Contributions:**

This paper proposes a novel context-aware OOD intent detection (Caro) framework for multi-turn dialogue contexts, by tackling two main challenges: (1) How to alleviate the long distance obstacle and learn robust representations? (2) How to utilize unlabeled data? For the first challenge, by following the information bottleneck principle, Caro extracts robust representations from these contexts and removes irrelevant information using a multi-view information bottleneck loss. For the second one, Caro introduces a two-stage  self-training scheme to mine OOD samples. Specificalluy, the first stage builds a preliminary OOD detector with OOD samples synthesized from IND data, while the second stage refine it based on the real-world OOD samples selected from the unlabeled data. The author have conducted extensive experiments and analysis, showing that Caro is effective in OOD intent detection for multi-turn dialogue.

**Questions For The Authors:**

1. Which feature is used to generate the pesudo OOD samples in the first stage, the aggregated features or the single-view features?
2. What is the difference between the Adaptive Reception Field and the Attetion mechanism or convolution kernel?
3. Is the performance of the OOD intent detection positively correlated with the turns of the dialogue? Is the longer the round of the dialogue, the better the OOD detection? Can you give a fine-grained analysis?

**Reasons To Accept:**

1.The proposed method is clear and simple; It extracts robust intent representations from multi-turn dialogue contexts by combining the features from diverse views following the information bottleneck principle, which is novel and intuitive.
2.The analysis and experiments are somewhat comprehensive. The performance on multi-turn OOD detection tasks is quite impressive, with a significant improvement in F1-OOD score of over 29% compared to the previous top-performing method.
3.The task is interesting and the paper is well written.

**Reasons To Reject:**

1.There is a lack of theoretical analysis of the feasibility of the information bottleneck principle in OOD intent detection from multi-turn dialogue contexts, especially why it is more suitable for the dialogues with longer turns.
2.The proposed method has not been verified on other datasets with different distributions/domains, and it only focused on STAR with two versions (small/large), thus the universality needs to be further verified.

**Reproducibility:**

4: Could mostly reproduce the results, but there may be some variation because of sample variance or minor variations in their interpretation of the protocol or method.

**Reviewer Confidence:**

3: Pretty sure, but there's a chance I missed something. Although I have a good feel for this area in general, I did not carefully check the paper's details, e.g., the math, experimental design, or novelty.

---

> ### Author Rebuttal · Authors · 2023-08-28
>
> Thank you for your attentive comments! We are glad you thought "method is clear and simple", "analysis and experiments are comprehensive", and "paper is well written". We address your feedback point by point below.
>
> -- "There is a lack of theoretical analysis of the feasibility of the information bottleneck principle in OOD intent detection from multi-turn dialogue contexts, especially why it is more suitable for the dialogues with longer turns."
>
> 1. We thought the learned representations may contain superfluous information that is irrelevant for OOD intent detection tasks. We attempt to remove superfluous information by introducing multi-view information, because each view provides the same task-relevant information (Zhao-et-al.,2017). Hence, one can improve generalization by discarding all the information not shared by both views from the representation (Federici-et-al.,ICLR2019). We do this by maximizing the mutual information between the representations of the two views while at the same time eliminating the information not shared between them.
> 2. We also perform fine-grained analysis of the learned representations following (Tishby-et-al.,2000) in Figure 2. It can be seen that the information bottleneck loss helps Caro to learn ideal representation, which would be maximally predictive about the label while retaining a minimal amount of information from the observations.
>
> --"The proposed method has not been verified on other datasets with different distributions/domains, and it only focused on STAR with two versions (small/large), thus the universality needs to be further verified."
>
> 1. Yes, our method is mainly evaluated in a multi-turn dialogue dataset, STAR, which is designed to model long context dependence with explicit OOD intent annotations. Given existing OOD intent detection approaches often focus on single-turn interactions, it is really difficult for us to find additional datasets with muti-turn dialogue contexts and annotations of OOD intent. Still, we curate two variants of the STAR dataset (STAR-Full and STAR-Small) to evaluate the generalization of our method.
>
> --"Which feature is used to generate the pesudo OOD samples in the first stage, the aggregated features or the single-view features?"
>
> 1. The aggregated features are used to generate the pesudo OOD samples in the first stage.
>
> --"What is the difference between the Adaptive Reception Field and the Attetion mechanism"
>
> 1. Adaptive Reception Field is inspired by the observation in the neuroscience community that human brains process information with multiple receptive fields (Sceniak-et-al.,1999), where the receptive field size for neurons is adapted based on input stimuli (Spillmann-et-al.,2015) so that different regions of inputs are emphasized (Pettet-et-al.,1992). Attetion mechanism is a mechanism mimicking cognitive attention in the machine learning community, which calculates "soft" weights for each embedding in the context window.
>
> --"Is the performance of the OOD intent detection positively correlated with the turns of the dialogue? Is the longer the round of the dialogue, the better the OOD detection? Can you give a fine-grained analysis?"
>
> 1. Kindly note that OOD detection performances under different context lengths is analysed and shown in Table 5, in which we validate longer the round of the dialogue leads to better OOD detection performance.

---

### Official Review · Reviewer_j6vP · 2023-08-04

**Soundness:** 3

**Excitement:**

3: Ambivalent: It has merits (e.g., it reports state-of-the-art results, the idea is nice), but there are key weaknesses (e.g., it describes incremental work), and it can significantly benefit from another round of revision. However, I won't object to accepting it if my co-reviewers champion it.

**Paper Topic And Main Contributions:**

This paper introduces a novel framework called Caro for addressing the challenging and relatively unexplored problem of Out-of-Domain (OOD) intent detection in the context of multi-turn dialogues. Traditional OOD intent detection approaches often focus on single-turn interactions, but real-world dialogue systems often involve multi-turn conversations. Caro aims to accurately detect OOD intents while considering the complexities of multi-turn contexts.

**Reasons To Accept:**

1) The paper introduces a novel framework called "Caro" that addresses a challenging and under-explored problem in the field of intent detection.
2) By constructing diverse views of input data and optimizing an unsupervised multi-view loss, Caro retains predictive information relevant to intent detection while discarding irrelevant information.
3)Caro introduces a two-stage self-training process to mine OOD samples from unlabeled data. This addresses the challenge of refining OOD detection without access to labeled OOD samples during training.

**Reasons To Reject:**

I'm not very familiar with this area anymore. please getting the opinions of other reviewers.

**Reproducibility:**

4: Could mostly reproduce the results, but there may be some variation because of sample variance or minor variations in their interpretation of the protocol or method.

**Reviewer Confidence:**

2: Willing to defend my evaluation, but it is fairly likely that I missed some details, didn't understand some central points, or can't be sure about the novelty of the work.

---

> ### Author Rebuttal · Authors · 2023-08-28
>
> Thank you for the positive feedback！ We are glad you find it to be a "challenging and under-explored problem", and noted the empirical benefits of considering multi-turn dialogue contexts.

---

### Meta-Review · Area_Chair_22n5 · 2023-09-18

**Recommendation:** 3

**Metareview:**

In this work, the authors present Caro, a context aware out of domain intent detector that models multi turn dialog contexts, using a multiview approach to remove information not related to intent detection. The authors report very strong results, accompanied by comprehensive experiments, on a pair of variants on the STAR dataset.

The paper is reasonably well-written, but would be improved if the authors incorporated the notes from the discussion into the manuscript, since some aspects of methodology could be made clearer. The authors did a good job of addressing these questions in their rebuttal.

The core contributions of this paper is centered around a key insight: the use of the multiview approach to improve performance for this task, which is significant and worth improving upon, and reporting a clear and comprehensive set of experiments on the STAR dataset, which is appropriate for the task.

The main weakness is that the paper provides results on variants from a single dataset, leaving open the question of whether these improvements are generalizable across datasets. The author's comments on this topic are well-taken, but it is nonetheless a limitation: datasets that would be useful for this aren't available today, but that also provides an opportunity for contributions in this space.

As such, I think my recommendation below largely reflects the position of the reviewers.

---

### Decision · Program_Chairs · 2023-10-07

**Decision:**

Reject

**Comment:**

In this work, the authors present Caro, a context aware out of domain intent detector that models multi turn dialog contexts, using a multiview approach to remove information not related to intent detection. The authors report very strong results, accompanied by comprehensive experiments, on a pair of variants on the STAR dataset.

The paper is reasonably well-written, but would be improved if the authors incorporated the notes from the discussion into the manuscript, since some aspects of methodology could be made clearer. The authors did a good job of addressing these questions in their rebuttal.

The core contributions of this paper is centered around a key insight: the use of the multiview approach to improve performance for this task, which is significant and worth improving upon, and reporting a clear and comprehensive set of experiments on the STAR dataset, which is appropriate for the task.

The main weakness is that the paper provides results on variants from a single dataset, leaving open the question of whether these improvements are generalizable across datasets. The author's comments on this topic are well-taken, but it is nonetheless a limitation: datasets that would be useful for this aren't available today, but that also provides an opportunity for contributions in this space.

As such, I think my recommendation below largely reflects the position of the reviewers.